# Visceral Dissemination of Mucocutaneous Leishmaniasis in a Kidney Transplant Recipient

**DOI:** 10.3390/pathogens10010018

**Published:** 2020-12-30

**Authors:** Nídia Marques, Manuela Bustorff, Anabela Cordeiro Da Silva, Ana Isabel Pinto, Nuno Santarém, Filipa Ferreira, Ana Nunes, Ana Cerqueira, Ana Rocha, Inês Ferreira, Isabel Tavares, Joana Santos, Elsa Fonseca, Conceição Moura, André Cerejeira, Júlia Vide, Jorge Cancela, Joana Sobrinho Simões, Susana Sampaio

**Affiliations:** 1Kidney Transplant Department, Centro Hospitalar e Universitário de São João, Faculdade de Medicina da Universidade do Porto, 4200-319 Oporto, Portugal; maria.guerra@chsj.min-saude.pt (M.B.); ana.silva.ferreira@chsj.min-saude.pt (F.F.); ateresa.nunes@chsj.min-saude.pt (A.N.); u010279@chsj.min-saude.pt (A.C.); u012439@chsj.min-saude.pt (A.R.); inescastroferreira@sapo.pt (I.F.); isabel.salome@chsj.min-saude.pt (I.T.); u001835@chsj.min-saude.pt (J.S.); susana.sampaio@sapo.pt (S.S.); 2Parasite Disease Group, Instituto de Biologia Molecular e Celular, Instituto de Investigação e Inovação em Saúde da Universidade do Porto, i3S, 4200-135 Oporto, Portugal; cordeiro@ibmc.up.pt (A.C.D.S.); anaisabelpinto@ibmc.up.pt (A.I.P.); santarem@ibmc.up.pt (N.S.); 3Biological Science Department, Faculdade de Farmácia, Universidade do Porto, 4050-313 Oporto, Portugal; 4Pathological Anatomy Department, Centro Hospitalar e Universitário de São João, Faculdade de Medicina da Universidade do Porto, 4200-319 Oporto, Portugal; efonseca@ipatimup.pt (E.F.); u030973@chsj.min-saude.pt (C.M.); 5Dermatology and Venereology Department, Centro Hospitalar e Universitário de São João, Faculdade de Medicina da Universidade do Porto, 4200-319 Oporto, Portugal; andre.cerejeira@chsj.min-saude.pt (A.C.); u010822@chsj.min-saude.pt (J.V.); 6Hematology Department, Centro Hospitalar e Universitário de São João, Faculdade de Medicina da Universidade do Porto, 4200-319 Oporto, Portugal; jorge.cancelapires@chsj.min-saude.pt; 7Clinical Pathology Department, Centro Hospitalar e Universitário de São João, Faculdade de Medicina da Universidade do Porto, 4200-319 Oporto, Portugal; jssimoes@chsj.min-saude.pt

**Keywords:** leishmaniasis, transplant, immunosuppression

## Abstract

Intracellular protozoan of the genus *Leishmania*, endemic in the Mediterranean basin, are the cause of cutaneous (CL), mucocutaneous (MCL), and visceral leishmaniasis (VL). A 75-year-old woman was admitted nine years after a second kidney transplant (KT), due to persistent pancytopenia and fever. She presented edema and erythema of the nose in the last two years and an exophytic nodular lesion located on the left arm, with areas of peripheral necrosis and central ulceration in the last 18 months. A bone marrow biopsy revealed features compatible with *Leishmania* amastigotes, and polymerase chain reaction test (PCR) for *Leishmania infantum* was positive. Moreover, biopsy and PCR for *L. infantum* of the cutaneous lesion on the patient’s left arm and nose and PCR from peripheral blood were positive. Thus, a diagnosis of CL, MCL, and VL was made, and liposomal amphotericin B was initiated, but the patient had an unfavorable outcome and died. This is the first report of a KT recipient presenting with the entire spectrum of leishmaniasis. In Portugal, this infection is rare—so a high degree of clinical suspicion is required for its diagnosis, especially in endemic regions, as visceral leishmaniasis is a potentially life-threatening infection.

## 1. Introduction

Leishmaniasis is a chronic infectious disease caused by a complex of obligate intracellular protozoan of the genus *Leishmania*, is transmitted through the bite of infected female sandflies [1,2,3,4]. Clinical manifestations of the disease are mostly based on three different syndromes: Visceral leishmaniasis (VL), cutaneous (CL), or mucocutaneous (MCL) [1,2,3,5]. Leishmaniasis in the Mediterranean basin, is a zoonotic disease mostly caused by *Leishmania infantum*, usually associated with VL and less frequently with CL [6]. In 2009 there had been an outbreak of leishmaniasis in Madrid (Spain), with more than 700 cases declared, mainly presenting with the cutaneous form of the disease [7]. In Portugal, Lisbon metropolitan and Alto Douro regions are considered to be endemic for this disease [8]. The reported prevalence of CanL is 19.2% for Lisbon metropolitan area, and higher than 50% in the Alto Douro region, which represents a high risk of infection if we consider Canine Leishmaniasis (CanL), caused by *L. infantum*, an accurate measure of risk exposure to the parasite [9]. Given the high percentage of underreporting of VL in Portugal, it is difficult to estimate the impact of this disease on public health. There is only one study that characterized the epidemiological profile of Leishmaniasis in Portugal between 1999–2009, that reported 378 diagnosed cases of VL and estimated that less than 40% were notified to the Portuguese Directorate-General for Health (DGS) [9]. The last official report from DGS declares 35 VL in Portugal between 2013 and 2016 [10].

Although considered a rare disease in Portugal, the incidence in immunocompromised patients, like transplant recipients, a particularly vulnerable group being at higher risk of developing leishmaniasis, is increasing [11]. In immunocompetent patients, the principal manifestation is CL, but in solid organ transplant (SOT) recipients, CL is rarely described, and most published case reports refer to VL, with a prevalence among SOT of about 0.1% in endemic areas [4,11,12]. It’s estimated that the risk of VL in SOT can be 4 to 135 times superior to in immunocompetent patients [5,11,13]. Kidney transplant patients constitute most of the reported cases in the literature. In the absence of prospective studies to evaluate the incidence of the disease among SOT, it is not known if this is due to the greater number of kidney transplants in relation to other organs or to factors related to renal failure or dialysis [14,15,16]. Visceral leishmaniasis is fatal in over 95% of cases if not treated—thus, VL control critically depends on early diagnosis. The clinical presentation of VL in immunosuppressed individuals is often atypical and can be misdiagnosed or enshrouded by other comorbidities—thus, the diagnosis in transplant recipients may be delayed for several months, due to a low index of suspicion [17].

Herein, we report a case of a KT recipient that, nine years after transplantation, presented with simultaneously visceral, cutaneous, and mucocutaneous leishmaniasis, and discuss the challenges associated with diagnosis and treatment of this disease in KT patients. To our knowledge, this is the first report of a KT recipient presenting with the entire spectrum of the disease.

## 2. Case Presentation

A 75-year-old woman, residing in Aveiro district (traditionally a non-endemic area for leishmaniasis), often visited Alto Douro Region, a northern region of the country with a high rate of CanL. The patient had end-stage kidney disease of unknown etiology and was submitted to a second deceased donor KT in 2010, whose induction immunosuppression was performed with timoglubulin and maintenance immunosuppression with prednisolone, tacrolimus, and mycophenolate mofetil (MMF). Following the second transplant, the patient was on low dose prednisolone (5 mg), tacrolimus (through levels 4–7 ng/mL), and MMF 500 mg per day. Nine years later, the patient was admitted due to fever, the presence of vesicles, and ulcerations in the oral mucosa and palate, and pancytopenia. Pancytopenia led to MMF withdrawal one week before admission, due to its myelosuppressive side effect [18]. The oral lesions were treated successfully with acyclovir, due to suspicion of an oral herpes virus infection, which was confirmed by a positive PCR test result to herpes simplex virus type 1, on a swab sample. Additionally, she complained of anorexia, asthenia, and a progressive weight loss of 22% in the past 12 months. At her physical examination, she was pale, looked malnourished, and was without lymph node enlargement or palpable hepatosplenomegaly. The patient presented edema and erythema of the nose, in the last two years, presumably resulting from a sulfuric acid burn many years before. She also had an exophytic nodular lesion located on the left arm, with areas of peripheral necrosis and central ulceration in the last 18 months, thought to be a cutaneous neoplasm, but missed surgical removal appointments. She recently noticed a progressive growth of the lesion (Figure 1A and Figure 2A). Her lab results, beyond pancytopenia (hemoglobin 9.8 g/dL, white blood cells 1310/mm^3^, neutrophils 750/mm^3^, and platelets 41/mm^3^) revealed albumin 29.5 g/L, ferritin 3194.9 mg/dL, and on protein electrophoresis alpha-1, alpha-2, beta, and gamma globulins were within the normal range. Additionally, there were severe alterations on coagulation tests, with a prothrombin time of 18.9 s (normal range 9.9–13.6 s), an activated partial thromboplastin time of 48.9 s (normal range 24.2–36.4 s), a fibrinogen of 60 mg/dL (normal range 200–400 mg/dL) and D-dimers of 12.08 µg/mL (normal range 0–0.5 µg/mL).

The patient evolved with persistent pancytopenia and fever of unknown etiology. Blood and urine cultures were negative, as well as PCR for herpes simplex virus, cytomegalovirus, parvovirus B19, typical and atypical mycobacteria. PCR for Epstein Barr was positive (8.9 × 10^4^ copies/mL, 4.9 log). A cervical, chest, and abdomen computed tomography scan showed only homogenous hepatomegaly, without lymph node enlargement, splenomegaly, or other relevant alterations. A bone marrow (BM) biopsy was performed, ruling out hemophagocytosis and revealing features compatible with Leishmania amastigotes (Figure 3).

The seroreactivity towards *Leishmania* antigens was assessed by immunofluorescence antibody test (IFAT) and Enzyme-Linked Immunosorbent Assay (ELISA), reference tests for Leishmaniasis diagnosis [19]. IFAT is based on the detection of Abs against *Leishmania* promastigotes and is often used in clinical settings [19]. IFAT was performed using the slides coated with promastigotes of *L. infantum* (zymodeme MON-1) from Fluoleish Kit (BioVeto), according to the manufacturer’s instructions. Antibody binding was revealed using anti-human IgG Alexa 488 (ThermoFisher, A-11013) as a secondary antibody, and fluorescence was analyzed in ZOE Fluorescent Cell Imager (Bio-Rad). As a positive control, plasma from a patient with a confirmed diagnosis for visceral leishmaniasis was used and as a negative control, a test with the secondary antibody only was included. The patient’s plasma sample at a 1:160 dilution was positive for the presence of anti-*Leishmania* IgG antibodies (positive IFAT ≥ 1:80).

The presence of *Leishmania*-specific antibodies in the plasma was also confirmed by ELISA assay [20]. Briefly, 96-well flat-bottomed microtiter plates (Greiner, Frickenhausen, Germany) were coated with 10 μg/mL of soluble promastigote *Leishmania* antigens (SPLA) or 5 μg/mL of rK28. Plates were incubated overnight at 4 °C and blocked with 200 μL of PBS-low-fat-milk 10% at 37 °C for 1 h. Plates were then washed with PBS-Tween 0.05% (PBS-T), and the plasma, positive and negative controls diluted 1:400 in PBS-T-low-fat-milk 1%, were dispensed in duplicate (100 μL/well) and incubated at 37 °C for 1h. Anti-human IgG conjugated to horseradish peroxidase (ref. A8667; Sigma, St. Louis, MO, USA) was used as a secondary antibody diluted at 1:5000, in accordance with the titration performed for the lot, and added to the plates were incubated at 37 °C for 30 min. Plates were washed and incubated with 0.5 mg/mL of o-phenylenediamine dihydrochloride (Sigma) for 10 min in the dark. Then the reaction was stopped by the addition of 50 µL/well of HCl 3 M. The absorbance at 492 nm was read in an automatic plate reader (Synergy 2, BioTek Instruments, Inc., Winooski, VT, USA). The sample and antigens were assayed in duplicate in two independent assays.

For the ELISA in which a promastigote lysate served as the antigen source, SPLA was prepared in-house as described previously [21]. The detection of Abs against SPLA was shown to be sensitive and specific [22]. Nonetheless, SPLA can be associated with some cross-reactivity, due to the existence of epitopes that might be conserved in other pathologies, such as tuberculosis [23]. Recombinant antigens like the protein rK28 (or rK39) are associated with increased specificity while maintaining comparable sensitivity to SPLA [20,24,25,26].

The recombinant fusion protein rK28 (kinesin-related protein of *L. infantum* rK39 flanked by the repeat sequences of HASPB1 and the ORF of HASPB2) in ELISA displays performance comparable or superior to those observed for rK39, the antigen most commonly available commercial antigen [19,27]. rK28 has shown promising results similar to rK39 and offers an additional tool in the antigen arsenal for VL diagnosis [24,25]. The optical density (OD) of the sample was above the cut-off for both antigens, being, therefore, considered seropositive. As a positive control, the serum of a patient with a confirmed diagnosis of Leishmaniasis was used. The absolute mean values of two independent tests are shown in Table 1.

A PCR test for *Leishmania* made in peripheral blood and in bone marrow (BM) aspirate came out positive. A biopsy of the cutaneous lesion in the patient’s left arm revealed squamous cell carcinoma and no amastigotes (Figure 4a). Despite presumably caused by a chemical burn, a biopsy of the patient’s nose was obtained and reveal features of *Leishmania* amastigotes (Figure 4b). A PCR test for *Leishmania* done in the nose and left arm biopsies, was also positive.

PCR-based methods have been used for typing *Leishmania* parasites, among which the ribosomal DNA internal transcribed spacer-1 (ITS) region has been extensively validated given its high resolution for *Leishmania* discrimination in the Old World [28]. As such, total genomic DNA (gDNA) extracted from the patient´s BM and cutaneous lesions, together with a reference strain of *L. infantum* (MHOM/MA/67/ITMAP-263), were used to perform molecular typing by sequencing the PCR product generated using ITS-1 specific primers and then performing sequence alignments against the *Leishmania* genome database, as described previously [29]. Given the nature of the clinical manifestations of leishmaniasis, the products from each biological specimen were sequenced independently. Sequences recovered from the patient´s BM and cutaneous lesion showed 100% of homology for ITS-1 sequence amongst themselves (data not shown). The patient´s consensus sequence was then compared with the laboratory reference strain for *L. infantum* (MHOM/MA/67/ITMAP-263) and additionally with the reference strains for *L. infantum* (MHOM/PT/00/IMT260), for *L. donovani* (MHOM/CN/58/Sman), for *L. major* (MHOM/AT/16/AS16) and for *L. tropica* (MHOM/KE/84/NLB297), all sequences were retrieved from the National Center for Biotechnology Information (NCBI) GenBank. The patient´s consensus sequence only displayed 100% of homology for the ITS-1 sequence with *L. infantum* strains—thus, the ITS-1 DNA sequences obtained from the patient were consistent with *L. infantum* species (Figure 5 and Table 2).

A diagnosis of CL, MCL, and VL was made, and intravenous liposomal amphotericin B therapy was started at 4 mg/kg per day for five days and then continued for five more weeks (4 mg/kg per week). Besides, tacrolimus dose was reduced (though level 2–4 ng/mL), and MMF was already suspended. There was a rapid improvement of cutaneous and mucocutaneous lesions (Figure 1B and Figure 2B), however, pancytopenia persisted. Quantitative PCR obtained 18 and 28 days after treatment initiation was 7.1 × 10^5^ pg and 1.7 × 10^4^ pg of *Leishmania* DNA/mL whole blood, respectively. However, the patient evolved unfavorably. She developed bacteriemia, due to *Pseudomonas aeruginosa* and *Serratia marcescens*, with poor response to treatment and progressive cachexia. As the patient was very severely frail, after discussion with family, palliative care was offered. The patient died shortly after.

## 3. Discussion

In SOT, leishmaniasis can be a de novo infection, acquired through the bite of the vector, reactivation of a latent infection induced by immunosuppression, or an infection derived from the organ donor, which should probably be the less frequent mechanism of infection [1,4,12,17]. Screening for leishmaniasis in donors and candidates is not part of routine pre-transplant evaluation protocol in most kidney transplantation centers, including in Portugal, so it is difficult to determine the route of infection in the majority of the cases [2]. The incidence of VL is increasing due to several converging factors, including extensive traveling and migrations between endemic and non-endemic areas; therefore, serological screening for this agent should be included in pre-transplant evaluation and donor screening protocols. The infection can have a long period of incubation, and symptoms may only manifest several years after infection [30]. Some patients may remain asymptomatic, even when immunocompromised. In a Spanish study, the prevalence of asymptomatic *Leishmania* infection by serological technique was 4.8%, but other series report up to 30% of asymptomatic patients [31]. Leishmaniasis usually presents with a medium delay of 19 months after KT, although considering all SOT, the medium delay is about 11 months [1,5,11,32]. There is a paucity of information regarding risk factors among SOT for the development of leishmaniasis. Old males seem to be the most affected [11,17,33], and in one multicenter case–control study, the usage of high doses of steroids was the only risk factor identified for VL [1,12]. Immunosuppression impairs T cell mediated immunity, critical to controlling intracellular organisms like leishmania protozoal, and increases the risk of clinical or more severe disease [2,12,17]. Although our patient was not treated with high dose steroids, the immunosuppressed condition probably contributed to the severe course of her disease. Leishmaniasis was diagnosed nine years after kidney transplantation, and she traveled frequently to an endemic region. The long period of time since transplantation to the appearance of the muco-cutaneous lesions, suggests that she probably had a primary infection, rather than a reactivation or a donor-derived infection, although we cannot completely dismiss these two later possibilities.

In immunocompetent patients, CL is the principal clinical syndrome of leishmaniasis, however, in immunocompromised patients, CL and MCL are seldom described in case reports. CL can occur before, simultaneously, or after visceral involvement, called post-kala-azar (VL) dermal leishmaniasis, mostly described in immunocompromised patients [4,34]. VL may be the primary manifestation of leishmaniasis, and can represent the dissemination of the parasites through the reticuloendothelial system and visceralization of a primary cutaneous lesion. The opposite may also occur, as an initial visceral infection can disseminate and cause a secondary CL [4,35]. Skin lesions may present with a variety of phenotypes, making diagnosis difficult, as it can be misdiagnosed as malignancy or opportunistic infection, as in the case of our patient. VL is the most frequently reported manifestation of leishmaniasis in SOT, with frequently late diagnosis, due to the low index of suspicion or misdiagnosis [17]. Classic features include fever (the most report symptom in SOT), pancytopenia, and organomegaly (less frequent in SOT than in immunocompetent patients) [4,17]. This triad is only present in a third of the cases [1,4]. Our patient presented these features (fever, pancytopenia, and hepatomegaly); however, a low index of suspicion, in part due to scarcity of human leishmaniasis in Portugal, led to late diagnosis. MCL in Portugal is extremely rare and not frequently associated with *L. infantum*. Simultaneous VL, CL, and MCL were described in HIV and other immunosuppressed patients, but to the best of our knowledge, this is the first report of a KT recipient presenting with the entire spectrum of the disease [6]. We believe that the first manifestation of leishmaniasis in our patient was CL and MCL with posterior dissemination through the reticuloendothelial system and visceralization of parasites, facilitated by immunosuppression and the long course of the disease before diagnosis was made. Secondary infections should always be suspected in case of new-onset fever, since they can occur in about 22% of patients with VL, often reflecting a high degree of immunosuppression [1,17,35]. Activation of the coagulation cascade with hypofibrinogenemia and high D-dimers levels were present in our patient, as has also been described in a few case reports of VL [36]. Increased production of tissue factor related to the immune response induced by VL seems to be involved in consumptive coagulopathy. Blockage of the mononuclear phagocytic system, due to infection of macrophages and an increase in the production of interleukin-1 and tumor necrosis factor in the parasitized tissues, may contribute to the increase of tissue factor for coagulation [37,38]. In VL, polyclonal B cell activation with hypergammaglobulinemia and circulating autoantibodies are also described, and VL can be misdiagnosed as an autoimmune disease. Cryoglobulinemic diseases associated with VL were also described in some case reports [36,39]. Our patient didn’t show these features, with normal globulin levels and negative autoimmune test results. Other important differential diagnosis of VL that can be excluded in our patient, were other opportunistic infections, such as atypical mycobacterial infections, post-transplant lymphoproliferative disease and hemophagocytic syndrome, which can also be secondarily induced by leishmaniasis [4,36]. In endemic regions, disseminated histoplasmosis should also be considered in the differential diagnosis of VL [36].

The gold standard for diagnosis of VL is the visualization of amastigotes on direct microscopy of tissues using Wright-Giemsa stain, most commonly bone marrow or spleen aspirates [1,3,4,5]. This technic has high sensitivity, being up to 86%, but it’s an invasive procedure and doesn’t permit species identification [4]. Culture is possible, however, it is laborious and takes up to four weeks [1,4]. Serologic assays have better sensitivity in SOT than in HIV patients (92% vs 48%) [17], but do not differentiate between active from past infection [1,3,4,30]. Still, cross-reactivity with Chagas disease, tuberculosis, toxoplasmosis, malaria, typhoid fever, or brucellosis often results in false-positive serology for leishmaniasis, which is particularly relevant in patients more prone to comorbidities [40]. Therefore, in immunocompromised patients, for leishmaniasis is recommended to perform at least two serological assays in addition to the PCR [17]. In the present study, the diagnosis was established by serological and molecular approaches. The serological results were consistent with the presence of a robust *Leishmania* specific response detected both by IFAT and SPLA/k28 ELISA.

Molecular techniques are being increasingly used. *Leishmania* diagnosis by PCR has high sensitivity and specificity, and can also be used in different samples. In peripheral blood, PCR sensitivity is similar to microscopy examination of bone marrow aspirates for amastigotes detection in HIV patients [41]. This technique not only detects and quantifies *Leishmania* amastigotes, but also species identification [4,17]. PCR-based methods targeting different gene regions of the *Leishmania* genome followed by DNA sequencing are highly sensitive for the identification of *Leishmania* species in clinical samples, as shown in our case [28,42]. Quantitative PCR on peripheral blood may be useful to monitor therapy or to identify recurrence after treatment, and many authors recommend it to monitor response to therapy [17,43]. However, tests are not standardized, and no threshold value is defined to consider a positive PCR result as indicative of clinical disease, at least in immunocompromised patients [44,45]. Since it is a highly sensitive technique, a positive blood PCR does not indicate clinical disease and may persist many months after successful treatment [46]. CL and MCL diagnosis are based on the identification of parasites by direct microscopy or PCR of tissue obtained by biopsy or swab [1,4]. PCR is the most sensitive test, as shown in our case as biopsy of the lesion of the arm did not show the amastigotes, however, PCR test was positive.

Little is known regarding how to treat VL in SOT, and the current knowledge is mostly derived from experience in HIV patients. Guidelines from the American Society of Transplantation recommend, as a first step, reduction of immunosuppression and liposomal amphotericin B 4 mg/kg/day for five days, and then five cycles of once-a-week infusion, to achieve a cumulative dose of 40 mg/kg [1,3,36]. The resistance of *L. infantum* is seldom described. Clinical improvement is expected in the first five days and hematological improvement in two weeks [4]. In our patient, persistent pancytopenia after two weeks of treatment was observed. Quantitative PCR for *Leishmania* DNA decreased, however, it was not followed by clinical improvement. We presume that this may be due to slower treatment response in relation to the lounger course of the disease, a high burden of disease, associated cachexia, and a reflection of the high degree of immunosuppression. There are other treatment options, including pentavalent antimonial compounds, traditionally considered the first option for VL treatment. Nowadays, these agents are a second-line option, since they are associated with serious adverse effects like hepatitis and pancreatitis [4,39]. Miltefosine is an oral option, principally used for the treatment of VL caused by *L. donovani*, but data regarding its use in SOT is scarce, and with evidence of *L. infantum* resistance [1,4,47]. After completion of therapy, SOT should be monitored for relapse for a minimum of 12 months, as a high rate of recurrence is observed in these patients (about 30%) [1]. Contrary to patients with HIV/AIDS, secondary prophylaxis (chronic maintenance therapy) after successful treatment is not recommended in SOT [5,11].

Leishmaniasis is a rare disease in SOT. Besides, the clinical manifestations are not specific. These facts make the diagnosis challenging and requiring the PCR-based molecular assays that allow the diagnosis with non-invasive methods. Another significant point is the importance of being alert to the presence of chronic skin lesions in SOT, since they may be related to atypical infections, as reported in this case. Delayed diagnosis can lead to cachexia, treatment failure, and a fatal outcome.

## Figures and Tables

**Figure 1 pathogens-10-00018-f001:**
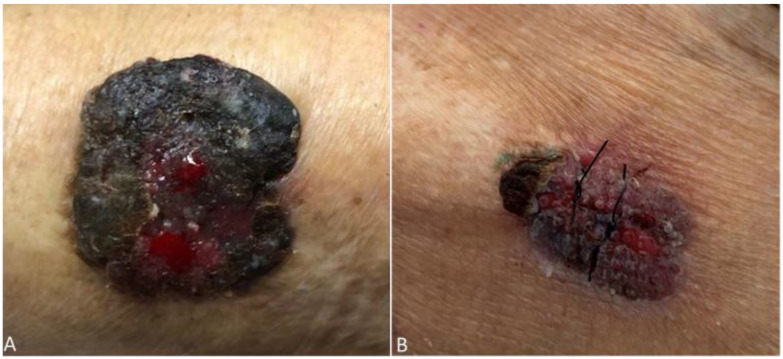
Ulcerative lesion on the left arm present in the past 18 months before (**A**) and after (**B**) initiation of Leishmaniasis treatment.

**Figure 2 pathogens-10-00018-f002:**
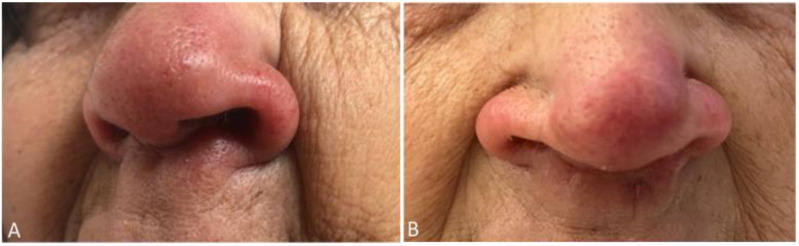
Patient’s nose lesion before (**A**) and after (**B**) initiation of Leishmaniasis treatment.

**Figure 3 pathogens-10-00018-f003:**
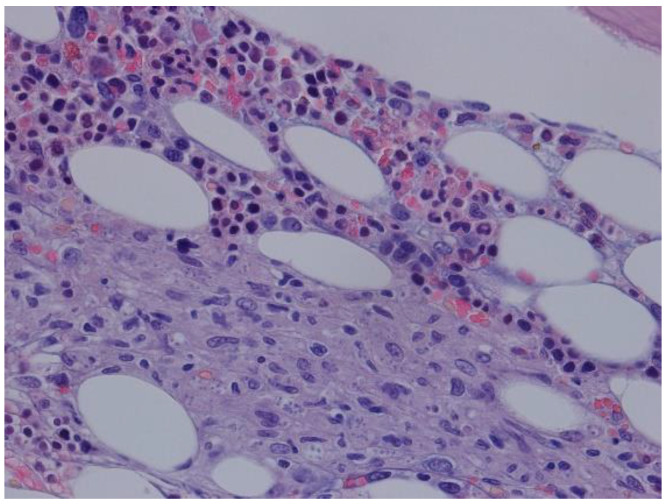
HE 400×—bone marrow with macrophage aggregates with structures compatible with *Leishmania*.

**Figure 4 pathogens-10-00018-f004:**
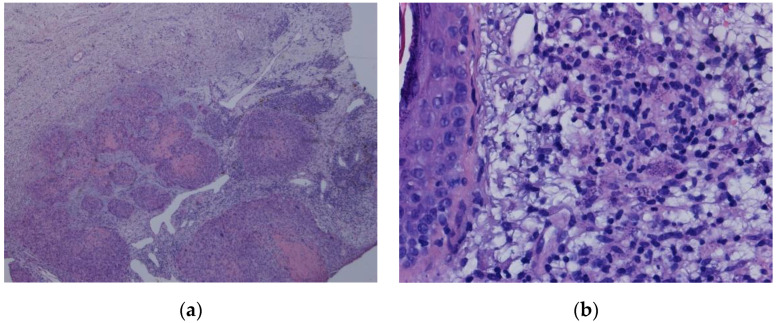
(**a**): HE 200×—squamous cell carcinoma; (**b**): HE 400X—skin with inflammatory infiltrate and macrophage with structures compatible with *Leishmania*.

**Figure 5 pathogens-10-00018-f005:**
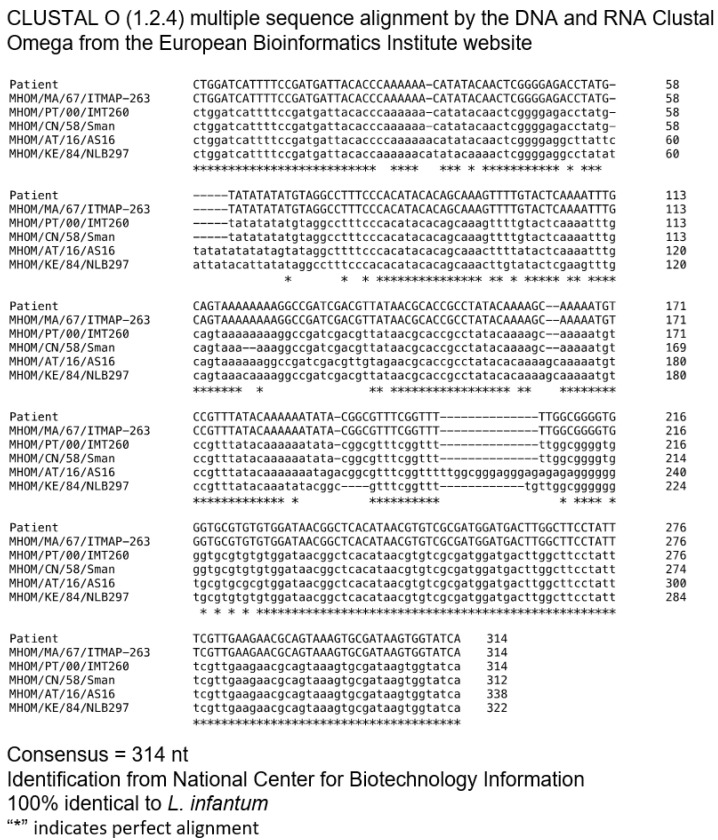
Multiple sequence alignment (Clustal Omega) of nucleotide sequences of ITS-1. Comparison of ITS-1 sequence obtained from a patient with ITS-1 sequences and the following reference strains of *L. infantum* (MHOM/MA/67/ITMAP-263), *L. infantum* (MH).

**Table 1 pathogens-10-00018-t001:** ELISA results expressed in optical density (OD).

	SPLA	Rk28
**Patient**	0.313	0.466
**Positive Control**	0.724	1.015
**Cut-off**	≥0.092	≥0.096

**Table 2 pathogens-10-00018-t002:** Percentage of homology ITS-1 Sequence.

	(*L. infantum*) MHOM/MA/6 7/ITMAP-263	(*L. infantum*) MHOM/PT/00 /ITM260	(*L. donovani*) MHOM/CN/58 /Sman	(*L. major*) MHOM/AT/16 /AS16	(*L. tropica*) MHOM/KE/84 /NLB297
Patient	100%	100%	99%	89%	91%

## Data Availability

Data sharing not applicable. No new data were created or analyzed in this study. Data sharing is not applicable to this article.

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
