# Peer review of "Visceral Dissemination of Mucocutaneous Leishmaniasis in a Kidney Transplant Recipient"

_pathogens, 2020, doi:10.3390/pathogens10010018_

Round 1
Reviewer 1 Report
line 27: What is a hypertrophic lesion in the arm ? muscle hypertrophy ? keratosis in the skin ? The microscopic picture of SCC was shown below. It is better to use medical term in dermatology.
line 32: evolution ? do you mean "outcome" ?
line 33-34: Why is "a high index of suspicion" necessary ?
line 47-48: I could understand the meaning. But it is better to rewrite.
line 55-57: It said that immuno-compromised host including transplantation and leishmaniasis are already reported. What is the new finding of this case report ?
line 59: What does 0.1 % represent for ? VL occurrence in compromised patients ?
line 60-62: What is the description bias ?
line 63-65: What is the typical misdiagnosis for VL in immuno-compromised patients ?
line 77: I assume that long term usage of steroid can induce pancytopenia.
line 88: What does "unremarkable serum globulin electrophoresis" mean ? The monoclonal or polyclonal pattern of gamma globulin increase ? The rate of alpha or beta globulin is changed ?
line 89-90: PT and APTT are within normal range ? fibrinoge is decreased ? What is your point to describe these? You need explanation for these numbers.
line 101-121: The description of serum test for leishmaniasis is not clear. What is the difference of rK28 and 39 and why did you use rK28 ? What is the difference of SPLA and rK28 antibodies ? Why did you test both ?
line 159-160: Why don't you show copy numbers since you are performing qPCR ?
line 161: do you mean "sepsis " ? or bacteremia ?
line 174: centers? Do you mean countries ? hospitals ?
line 174-175: mechanism of infection? Instead, it seems a "route of infection".
line 178: "wide" --> long ?
line 206-207: It is not clear what authors insist the importance in KT patients who got VL. Because of usage of immune suppression therapy or 10 years after transplant in which the donor organ was contaminated with the leishmania ? Anyway, it was not proved either of those contributed to VL.
line 220: The sentence is not clear. Do you mean, "which can be secondarily induced by leishmaniasis "?
line 225: What is 14 ?
line 236 How is the PCR similar to direct microscopy ? What do you mean as "direct" microscopy ? Fluorescent detection with antibody against leishmania antigen ?
line 241-242: Scientific explanation is needed for why qPCR data doesn't reflect to the successful treatment.
line 252: qPCR for what ? leishmania genome ?
line 251-252: Do you mean that the recovery from pancytopenia must be accompanied with the decrease of leishmania genome as a sign of improvement for leishmania therapy ? I assume the recovery from pancytopenia is not necessarily required for the treatment of leishmania.
line 252-254: It is not clear what authors insist with these explanations.
line 254-257: It is difficult to follow these parts. More explanation is necessary.
line 257: What is secondary prophylaxis ?
line 260 What is "challenge " ?
As a whole, more clear explanations are required. The fact that VL is induced in the immunosuppressed patient must be explained, based on the literature. Diagnostic theory, method, technique must be explained clearly.
Reviewer 2 Report
The authors submitted a manuscript reporting a clinical case of a 75-year-old female living in Portugal having a gone through a second kidney transplant 9 years before the reported medical events. Due to the transplant the patient under permanent immunosuppressive drugs avoid transplant rejection but also making the patient more susceptible to various infections. Portugal is described as an endemic country for leishmaniasis mostly due the zoonotic transmission of L. infantum, a species of Leishmania parasite that also infects human beings. The present case is atypical since the patient presents simultaneously with cutaneous, muco-cutaneous and visceral manifestations of the disease.
The authors conducted several diagnostic tests looking for viruses and bacteria to find the origin of the different symptoms. Two different serological tests to detect the presence of anti-Leishmania antibodies were performed as well as a bone marrow biopsy to visually confirm the presence of the parasite as it is the gold standard. PCR followed by sequencing identified the parasite as being L. infantum. The patient received the appropriate first line drug treatment, liposomal amphotericin B and showed to be responsive to the treatment since the parasite load, tested by qPCR, was decreasing.
The authors can not discriminate if the patient was infected from the organ donation, which has been described in the past, since donors in Portugal are not tested for Leishmania prior organ donation or by cyclical transmission.
I believe the diagnostic strategy to diagnose leishmaniasis in an atypical environment was perfectly respected. The conclusions are pertinent and fit the results of the diagnostic tests. Reports of such clinical cases are very important since climate change is gonna change the geographical repartition of vector-borne pathogens and medical professionals that were never exposed to those pathogens will need those information.
Few minor typographic changes:
Line 100, 101, 109, 110, 111, 112, 131, please italicised “Leishmania”
Enjoy the end of the year
Round 2
Reviewer 1 Report
Mostly improved.
Some English usage must be corrected.
Line 215: may better to write as follows " Although our patients were not treated with high dose steroids, immunosupreesed condition contributed....."
Line 245: "altered immune response" instead of immune change
Line 248: tissue factor for coagulation ?
Line 252: negative data for autoimmune-related products ?
Line 253: It is better to add " that can be " before "excluded"
Line 255: "induced"
Line 272: How did the reference conclude PCR sensitivity is similar to that of microscopy examination ?
Line 280: need "," after technique
line 288: 5 cycles of once a week injection ?
line 290: need "," after patient
line 299: .....treatment is not expected, observed as a high rate of ........"
line 302: Leishmania is a rare diseas in SOT. Besides, the clinical manifestations are not specific. These facts make the diagnosis challenging and requiring the PCR-based molecular assays that allow the diagnosis with non-invasive methods.
